# Targeting of Ubiquitin E3 Ligase RNF5 as a Novel Therapeutic Strategy in Neuroectodermal Tumors

**DOI:** 10.3390/cancers14071802

**Published:** 2022-04-01

**Authors:** Elisa Principi, Elvira Sondo, Giovanna Bianchi, Silvia Ravera, Martina Morini, Valeria Tomati, Cristina Pastorino, Federico Zara, Claudio Bruno, Alessandra Eva, Nicoletta Pedemonte, Lizzia Raffaghello

**Affiliations:** 1Center of Translational and Experimental Myology, IRCCS Istituto Giannina Gaslini, 16147 Genoa, Italy; elisaprincipi@gaslini.org (E.P.); claudiobruno@gaslini.org (C.B.); 2UOC Genetica Medica, IRCCS Istituto Giannina Gaslini, 16147 Genoa, Italy; elvirasondo@gaslini.org (E.S.); valeriatomati@gaslini.org (V.T.); federico.zara@unige.it (F.Z.); nicolettapedemonte@gaslini.org (N.P.); 3Stem Cell Laboratory and Cell Therapy Center, IRCCS Istituto Giannina Gaslini, 16147 Genoa, Italy; giovanna.bianchi1984@libero.it; 4Experimental Medicine Department, University of Genova, 16132 Genova, Italy; silvia.ravera@unige.it; 5Laboratory of Molecular Biology, IRCCS Istituto Giannina Gaslini, 16147 Genoa, Italy; martinamorini@gaslini.org (M.M.); alessandraeva@gaslini.org (A.E.); 6Department of Neurosciences, Rehabilitation, Ophthalmology, Genetics, Maternal and Child Health (DI-NOGMI), University of Genoa, 16132 Genoa, Italy; cristinapastorino22@gmail.com

**Keywords:** RNF5, neuroblastoma, melanoma, ubiquitin ligase, endoplasmic reticulum associated protein degradation

## Abstract

**Simple Summary:**

Neuroblastoma and melanoma represent two common aggressive tumors of infancy and adulthood, respectively, with the same origin and poor prognosis. Despite the aggressive treatment, advanced neuroblastoma and melanoma patients are often refractory to therapy, relapse and die. Thus, novel therapeutic strategies are urgently needed. In this study, we demonstrated that RNF5, a ubiquitin ligase involved in the degradation of misfolded proteins, was expressed in neuroblastoma and melanoma patients and positively correlated with better outcome. In line with this observation, Analog-1, a pharmacological activator of RNF5, exerted a potent cytotoxic effect on in vitro and in vivo neuroblastoma and melanoma models by modulating the metabolism, reducing the growth and inducing the death of tumor cells. This study is of high translational relevance since it validates RNF5 as an innovative drug target and supports the development of novel therapies for neuroblastoma and melanoma patients in order to ameliorate their clinical outcome.

**Abstract:**

RNF5, an endoplasmic reticulum (ER) E3 ubiquitin ligase, participates to the ER-associated protein degradation guaranteeing the protein homeostasis. Depending on tumor model tested, RNF5 exerts pro- or anti-tumor activity. The aim of this study was to elucidate the controversial role of RNF5 in neuroblastoma and melanoma, two neuroectodermal tumors of infancy and adulthood, respectively. RNF5 gene levels are evaluated in publicly available datasets reporting the gene expression profile of melanoma and neuroblastoma primary tumors at diagnosis. The therapeutic effect of Analog-1, an RNF5 pharmacological activator, was investigated on in vitro and in vivo neuroblastoma and melanoma models. In both neuroblastoma and melanoma patients the high expression of RNF5 correlated with a better prognostic outcome. Treatment of neuroblastoma and melanoma cell lines with Analog-1 reduced cell viability by impairing the glutamine availability and energy metabolism through inhibition of F_1_F_o_ ATP-synthase activity. This latter event led to a marked increase in oxidative stress, which, in turn, caused cell death. Similarly, neuroblastoma- and melanoma-bearing mice treated with Analog-1 showed a significant delay of tumor growth in comparison to those treated with vehicle only. These findings validate RNF5 as an innovative drug target and support the development of Analog-1 in early phase clinical trials for neuroblastoma and melanoma patients.

## 1. Introduction

Ubiquitination is a finely orchestrated process of degradation and turnover of cellular proteins whose deregulation is often associated to different pathological conditions including cancer [1]. Ubiquitination occurs as the result of the sequential actions of ubiquitin (Ub) activating enzyme (E1), Ub conjugating enzymes (E2s), and Ub protein ligases (E3s), ending with the retrotranslocation of the ubiquinated protein into the cytosol for degradation by proteasomes [2,3]. RNF5 (RING finger protein 5) is an endoplasmic reticulum (ER)-anchored ubiquitin E3 ligase and a component of the UBC6e-p97 complex which is implicated in ER-associated degradation (ERAD), a pathway involved in the maintenance of protein homeostasis [4]. RNF5 was originally defined as a regulator of cell motility via targeting paxillin ubiquitination, which alters its localization to focal adhesion [5]. More recently, RNF5 has been described as a quality control protein in the ER where it contributes to the clearance of misfolded proteins, including the F508del mutant cystic fibrosis (CF) transmembrane conductance regulator (CFTR) [6,7]. The inhibition of RNF5 by a novel drug-like small molecule, i.e., inh-2, caused significant F508del-CFTR rescue in immortalized as well as in primary bronchial epithelial cells derived from F508del homozygous CF patients [8]. In contrast, Analog-1, defined as a RNF5 activator based on its biological effects, abolished the activity of inh-2 on the same cellular models [8]. 

However, in cancer, the role of RNF5 is still controversial. Increased expression of RNF5 was observed in different malignancies including breast cancer, melanoma, hepatocellular carcinoma and acute myeloid leukemia (AML) where it correlated with poor prognosis and tumor progression [9,10,11]. In this connection, *rnf5^−/−^* mice presented alterations in the gut microbiota composition, which, in turn, contributed to increase the host antitumor immunity, limiting tumor progression [12]. In contrast to RNF5 pro-tumor activity, in breast cancer, RNF5 promoted ubiquitination and degradation of glutamine transporters SLC1A5 and SLC38A2, which were aberrantly folded following chemotherapy-induced ER stress. This event reduced tumor cell proliferation and increased autophagy and cell death [13]. Accordingly, high expression of RNF5 was associated with positive prognosis in breast cancer patients [13]. Furthermore, RNF5 was able to degrade phosphoglycerate dehydrogenase (PHGDH), the first enzyme of the serine synthesis pathway, which is significantly upregulated in many cancers [14]. Finally, high expression of RNF5 in patients with glioma was associated with an improved prognosis compared to patients with low expression [15]. Given the controversial pathophysiological role of RNF5 in cancer, we extended the study of its functional expression to neuroblastoma (NB) and melanoma, which represent two neuroectodermal tumors of infancy and adulthood, respectively, with the same embryonal origin and poor prognosis. The potential therapeutic effect of novel pharmacological modulators of RNF5 was also investigated on in vitro and in vivo experimental NB and melanoma models.

Here, we demonstrate that high RNF5 expression in both NB and melanoma patients correlates with a better prognostic outcome in terms of overall and event free survival. Using in vitro NB and melanoma models, we show that treatment with the RNF5 activator Analog-1 reduces tumor cell proliferation and viability, decreases the glutamine and glutamate intracellular level, impairs energetic metabolism by inhibition of F_1_F_o_ ATP-synthase (ATP-synthase) activity, and increases apoptosis by promotion of oxidative stress. The decrease in tumor cell proliferation and induction of cell death by Analog-1 persists in in vivo experimental models leading to reduced tumor growth. These findings validate RNF5 as a drug target for cancer, and strongly encourage further preclinical development of the RNF5 activator Analog-1 and possibly clinical trials in patients affected by NB and melanoma to address the safety and efficacy.

## 2. Materials and Methods

### 2.1. Gene Expression Analysis

Publicly available datasets found on the R2: Genomic Analysis and Visualization http://r2.amc.nl platform (accessed on 1 October 2021) were used for the evaluation of RNF5 expression in NB primary tumor and melanoma samples. The NB dataset (GSE49710) contains the expression profiles of 498 NB specimens (211 female and 287 male patients, with a diagnosis age ranging from 0 to 25 years; the 75% of patients received the diagnosis before 3 years of age) measured with the Illumina HiSeq 2000 RNAseq platform [16] whereas the melanoma dataset (GSE65904) reports the gene expression profiles of 214 melanoma tumor samples analyzed with Illumina Human HT-12V4.0 BeadChip arrays [17]. *RNF5* differential expression and survival analyses were performed with R2 platform tools, using the median value as a cut-off to define high or low gene expression. The differential expression of RNF5 between high-risk and low-risk NB patients was reported as Log_2_ fold change (FC) value and the significance was calculated with ANOVA test. Patient overall survival (OS) and event-free survival (EFS) were assessed by Kaplan-Meier curves. The separation significance of the survival curves was assessed with log-rank test adjusted by the Bonferroni method.

### 2.2. Cells and Culture Conditions

Human NB cell lines (SH-SY5Y, SK-N-SH, IMR-32, HTLA-230), kindly provided by the Laboratory of Experimental Therapies in Oncology, Istituto Giannina Gaslini, Genoa, Italy, were periodically tested for mycoplasma contamination by polymerase chain reaction (PCR) assay, characterized by cell proliferation and morphology evaluation, and authenticated by multiplex short-tandem repeat profiling by BMR Genomics (Padova, Italy). Human melanoma cell lines MZ2-MEL were purchased from Dr. T. Boon, Ludwig Institute for Cancer Research, Brussels, Belgium as reported [18]. Human melanoma cell lines A2058, G-361, SK-MEL-28 were kindly donated by the Laboratory of Biotherapies, IRCCS Ospedale Policlinico San Martino, Genoa, Italy. Normal human fibroblasts were purchased from American Type Culture Collection (ATCC, Manassas, VA, USA) as reported [19]. Human tumor cell lines and fibroblasts were cultured with RPMI 1640 medium (Euroclone, Milan, Italy, catalog number ECM200) with standard glucose levels (1 g/L) and supplemented with 10% heat-inactivated fetal calf serum (Biochrom, Berlin, Germany), 50 IU/mL sodium penicillin G (Euroclone, catalog number ECB3001), 50 μg/mL streptomycin sulfate (Euro Clone, catalog number ECB3001), and 2 mML-glutamine (Euroclone, catalog number ECB3000D), as previously described [20]. Human bronchial epithelial (BE) immortalized cells (CFBE41o-) stably expressing the halide-sensitive yellow fluorescent protein (HS-YFP) YFP-H148Q/I152L and F508del-CFTR were generated as previously described [21]. CFBE41o- were cultured in MEM (Sigma Aldrich, St. Louis, MI, USA, catalog number M4655) supplemented with 10% fetal calf serum, 2 mM L-glutamine, 100 U/mL penicillin, and 100 µg/mL streptomycin.

The methods for the isolation, culture, and differentiation of primary BE cells were previously detailed [22,23]. In brief, epithelial cells were obtained from mainstem human bronchi, derived from CF and non-CF individuals undergoing lung transplant.

Primary human BE cells from a non-CF patient (HBE121) and primary human BE cells from a CF subject (F508del/F508del) (HBE55) were used for this study. To obtain differentiated epithelia, the cells were seeded at high density on porous membranes (Snapwell inserts, Corning, Teuksbury, MA, USA). After 24 h, the serum-free medium was removed from both sides and, on the basolateral side only, replaced with Pneumacult ALI medium (StemCell Technologies, Cambridge, UK, catalog number 05021) and differentiation of cells (up to 16–18 days) was performed in air-liquid interface (ALI) condition. The collection of BE cells (supported by Fondazione per la Ricerca sulla Fibrosi Cistica through the “Servizio Colture Primarie”) and their study to investigate the mechanisms of transepithelial ion transport were specifically approved by the Ethics Committee of the Istituto Giannina Gaslini following the guidelines of the Italian Ministry of Health (registration number: ANTECER, 042-09/07/2018). Each patient provided informed consent to the study using a form that was also approved by the Ethics Committee. All the cells were maintained at 37 °C, under 5% CO_2_ and 95% air humidified incubator.

### 2.3. Cell Viability

SH-SY5Y, MZ2-MEL cells and human fibroblasts were seeded in 25 cm^2^ cell culture flasks (1.5 × 10^6^ cells) and, one day later, they were treated with 10 µM of Analog-1, Inh-2 or vehicle alone (Dimethyl sulfoxide [DMSO], Sigma Aldrich, catalog number D9170) (negative control) for 24 or 48 h. Then, the cells were harvested, and counted by Trypan blue (Sigma Aldrich, catalog number T6146) Assay using a hematocytometer. Data are expressed as percentage cell viability over control ± SD.

### 2.4. SiRNA Transfection

SH-SY5Y and MZ2-MEL cells (2 × 10^6^ for each condition) were reverse-transfected with 30 nM (final concentration) of Non-Targeting (NT) or RNF5-targeting siRNA (Stealth, Life Technologies, Carisbad, CA, USA catalog number 129901 assay ID HSSS155076) using lipofectamine 2000 as transfection agent. Twenty-four hours after transfection and plating, the medium was changed, and the cells were incubated at 37 °C for an additional 24 h, prior to processing the cells for biochemical assays.

### 2.5. Western Blot

The different cell lines were grown to confluence on 60-mm diameter dishes and, after 24 h, lysed in RIPA buffer containing a complete protease inhibitor (Roche, Monza, Italy, catalog number 11836153001). Whenever a pharmacological treatment was required, 24 h after plating, the cells were treated with the indicated compounds and, after an additional 24 h, the cells were lysed. Cell lysates were subjected to centrifugation at 15,000 g at 4 °C for 10 min. Supernatant protein concentration was calculated using the BCA assay (Euroclone, catalog number EMP014250) following the manufacturer’s instructions. Equal amounts of protein (30 μg) were separated onto gradient 4–20% Criterion TGX Precast gels (Bio-rad laboratories Inc. Milan, Italy, catalog number 5671023), then transferred to nitrocellulose membrane with Trans-Blot Turbo system (Bio-rad Laboratories Inc.) and analyzed by Western blotting. Proteins were detected using antibodies listed in the Method Antibodies section. The proteins were then visualized by chemiluminescence using the SuperSignal West Femto Substrate (Thermo Scientific, Monza, Italy catalog number 34094). Chemiluminescence was monitored using the Molecular Imager ChemiDoc XRS System. Images were analyzed with ImageJ software (National Institutes of Health). Bands were analyzed as Region-of-Interest (ROI), normalized against the calnexin loading control. The molecular weight of the proteins (based on the Precision Plus Protein WesternC Standards, Bio-rad Laboratories Inc.) were calculated using the band analysis program of the software Quantity one 4.6 of the Molecular Imager ChemiDoc XRS System.

### 2.6. Antibodies

The following antibodies were used: mouse monoclonal anti-RNF5 22B3 (Santa Cruz, Heidelberg, Germany, catalog number sc-81716), rabbit polyclonal anti-calnexin (Abcam, Cambridge, UK, catalog number 10286), horseradish peroxidase (HRP)-conjugated anti-mouse IgG (Abcam catalog number 97023), and HRP-conjugated anti-rabbit IgG (Thermo Scientific, catalog number 31460).

### 2.7. Glutamine and Glutamate Determination

SH-SY5Y and M2Z-MEL were seeded in 25 cm^2^ cell culture flasks at 1.5 × 10^6^ cells/flask. One day later, the cells were treated with 10 µM of Analog-1 or vehicle alone (DMSO). After 24 or 48 h, the cells were harvested, counted, and evaluated for intracellular glutamine and glutamate content by a spectrophotometric commercial kit (Sigma Aldrich, catalog number GLN1) according to the manufacturer’s instructions. Data are expressed as mean ± SD (*n* = 3).

### 2.8. Glycolytic Enzyme Assays

SH-SY5Y and M2Z-MEL were seeded in 25 cm^2^ cell culture flasks at 1.5 × 10^6^ cells/flask and treated with 10 µM of Analog-1 or vehicle alone (DMSO) for 24 and 48 h. The assays were conducted on 50 µg of total protein. The reaction mixtures used for the determination of each enzyme activity were prepared as follows [24]: Hexokinase (HK, EC 2.7.1.1): 100 mM Tris-HCl pH 7.4 (Sigma Aldrich, catalog number T5941), 5 mM MgCl_2_ (Sigma Aldrich catalog number M8266), 200 mM glucose (Sigma Aldrich, catalog number G8270), 1 mM ATP (Sigma Aldrich, catalog number A2383, 0.91 mM NADP(Sigma Aldrich, catalog number N5755) and 0.55 IU/mL of glucose 6 phosphate dehydrogenase (G6PD) (Sigma Aldrich, catalog number G6378). Phosphofructokinase (PFK, EC 2.7.1.11): 100 mMTris-HCl pH 7.4, 2 mM MgCl_2_, 5 mM KCl (Sigma Aldrich, catalog number P3911), 2 mM fructose 6 phosphate (Sigma Aldrich, catalog number F3627), 1 mM ATP, 0.5 mM phosphoenolpyruvate (PEP) (Sigma Aldrich, catalog number 10108294001), 0.2 mM NADH (Sigma Aldrich, catalog number 10128023001) and 2 IU/mL of pyruvate kinase plus lactate dehydrogenase. Pyruvate kinase (PK, EC 2.7.1.40): 100 mM Tris-HCl pH 7.6, 2.5 mM MgCl_2_, 10 mM KCl, 0.6 mM PEP, 0.2 mM NADH, 5 mM ADP (Sigma Aldrich, catalog number A2754) and 1 IU/mL of lactate dehydrogenase (Sigma Aldrich catalog number 427217-M). Lactate dehydrogenase (LDH, EC 1.1.1.27): 100 mM Tris-HCl pH 9, 5 mM pyruvate and 0.2 mM NADH [25]. Data are expressed as mean ± SD (*n* = 3, for each enzyme)

### 2.9. Bioluminescent Luciferase ATP Assay

To evaluate the ATP synthesis trough the F_1_-F_o_ ATP-synthase, 50 μg of SH-SY5Y and MZ2-MEL cell proteins were incubated in the appropriate buffer plus 5 mM pyruvate and 2.5 mM malate as described in [26] and ATP synthesis was then induced by the addition of 0.1 mM ADP. ATP synthesis was measured in a luminometer (GloMax^®^ 20/20n Luminometer, Promega Italia, Milano, Italy) by the luciferin/luciferase chemiluminescent method (luciferin/luciferase ATP bioluminescence assay kit CLSII, Roche, Basel, Switzerland, catalog number 11699695001) [27]. In all experiments, the ATP synthesis rate was followed for 2 min, every 30 s. Data are expressed as mean ± SD (*n* = 3).

### 2.10. Evaluation of ATP and AMP Levels

ATP and AMP were measured according to the enzyme coupling method, following NAD(P)/NAD(P)H reduction/oxidation at 340 nm. For the ATP quantification, the assay was conducted in a medium containing: 100 mM Tris-HCl, pH 7.4, 5 mM MgCl_2_, 50 mM glucose, 0.2 mM NADP. The assay started after the addition of 4 µg of purified hexokinase and 2 µg of glucose-6-phosphate dehydrogenase. AMP concentration was assayed using the following medium: 100 mM Tris-HCl, pH 7.4, 5 mM MgCl_2_, 10 mM PEP, 0.15 mM NADH, 0.2 mM ATP. The assay started after the addition of 4 µg of purified pyruvate kinase plus lactate dehydrogenase and 2 µg of adenylate kinase [28]. Data are expressed as mean ± SD (*n* = 3).

### 2.11. Evaluation of Cytosolic Reactive Oxygen Species Production

To detect intracellular reactive oxygen species (ROS) production, SH-SY5Y and M2Z-MEL cells, exposed to 10 µM of Analog-1 or vehicle, were stained with Hoechst 33342 (Sigma Aldrich, catalog number 14533) to visualize nuclei, and with the ROS-sensitive cell-permeant probe 2′,7′-dichlorodihydrofluorescein diacetate (1.5 μM, H2DCFDA, Invitrogen, Monza, Italy, catalog number D399). Upon cleavage of the acetate groups by intracellular esterases and oxidation by cytosolic ROS, the non-fluorescent H2DCFDA was converted to the highly fluorescent 2′,7′-dichlorofluorescein (DCF). Cytosolic ROS production was then monitored in living SH-SY5Y and M2Z-MEL cells for up to 180 min by time-lapse imaging of cells using the Opera Phenix high-content screening system (PerkinElmer, Milan, Italy). DCF signal was laser-excited at 480 nm and the emission wavelengths were collected between 505 and 535 nm. Data are expressed as the mean value of DFCDA intensity ± SEM (*n* = 4).

### 2.12. Evaluation of Proliferation and Apoptosis

SH-SY5Y and M2Z-MEL cells were plated at low density (5000 cell/well) on 96-well plates suitable for high-content imaging. After 6 h, the cells were treated with 10 μM of Analog-1 or vehicle alone (DMSO). Cell proliferation was monitored for 24, 48, 72, 96, and 144 h using the Opera Phenix high-content screening system by the analysis of the percentage of open area. Data were expressed as the mean of percentage open area of 5 different fields ± SEM.

Alternatively, to monitor cytotoxic effect of the test compound, SH-SY5Y and M2Z-MEL cells were plated (10,000 cells/well) and treated with vehicle alone (DMSO) or Analog-1 10 μM or proteasome inhibitor MG-132 (Sigma Aldrich, catalog number M7449) for 24 and 48 h. Then, the cells were counterstained with Hoechst 33342 and propidium iodide (Invitrogen, catalog number P1304MP) to visualize total cells and apoptotic cells, respectively, and imaged by using the Opera Phenix. Data are expressed as the mean of % apoptotic cells ± SEM, (*n* = 6).

### 2.13. Evaluation of Autophagic Vacuoles with Monodansylcadaverine

SH-SY5Y and MZ2-MEL cells were plated (50,000 cells/well) on good-quality clear-bottom 96-well black microplates suitable for high-content imaging (Corning). After 24 h, the cells were treated with 10 μM Analog-1 or DMSO (negative control) for 24 h and with Torin-1 (Tocris Bio-techne, Milan, Italy, catalog number 4247) (autophagy inducer) and SAR405 (MedChemExpress, NJ, USA, catalog number HY-12481) (autophagy inhibitor). Another 24 h later, the cells were washed and incubated with 50 µM monodansylcadaverine (MDC) (Sigma Aldrich, catalog number D4008) in Phosphate Buffer Saline (PBS, Euroclone, catalog number. ECB4004) at 37 °C for 10 min and then analyzed [29]. High-content imaging and data analysis were performed using an Opera Phenix. Wells were imaged in confocal mode, using a 40× water-immersion objective. MDC signal was laser-excited at 405 nm and the emission wavelengths were collected between 435 and 550 nm. Data analysis of MDC spot number was performed using the Harmony software (ver 4.5) of the Opera Phenix high-content system. Data are expressed as means of n° spot ± SEM (*n* = 3).

### 2.14. Analysis of Actin Cytoskeleton and Evaluation of Cell Morphology

SH-SY5Y and MZ2-MEL cells were plated (50,000 cells/well) on good-quality clear-bottom 96-well black microplates suitable for high-content imaging. After 24 h the cells were treated with 10 μM Analog-1 or DMSO (as negative control) for 48 h. Then, the cells were washed, formalin-fixed and actin was labeled using phalloidin conjugated to Alexa Fluor 647 (Abcam, catalog number 176759). Cell nuclei were counterstained with Hoechst 33342. High-content imaging was performed using an Opera Phenix. Wells were imaged in confocal mode, using a 40× water-immersion objective. Phalloidin signal was laser-excited at 640 nm and the emission wavelengths were collected between 650 and 760 nm. Excitation and emission wavelengths for visualization of Hoechst 33342 signal were 405 and between 435 and 480 nm, respectively.

Analysis of cell morphology, including cell roundness, was performed using the PhenoLogic methods, based on machine-learning, of the Harmony software (version 4.9, PerkinElmer, Waltham, MA) of the Opera Phenix high-content system. The analysis was based on object classification and evaluation of multiple properties, such as size, intensity, texture, and morphology, according to specific algorithms developed by PerkinElmer and included in the Harmony software. Data were expressed as mean of single cell roundness ± SEM (*n* = 3000).

### 2.15. In Vivo Studies

Five weeks old female Athymic Nude (nu^−^/nu^−^) mice (Envigo, Italy, S. Pietro al Natisone, Italy) were housed in the animal facility at Policlinico San Martino Genoa under standard specific pathogen–free conditions and allowed access to food and water ad libitum. All experimental protocols were approved by the Policlinico San Martino Animal Welfare Body and by the Italian Ministry of Health (Authorization n° 355/2020-PR 23 April 2020). The mice (*n* = 6 /group) were inoculated subcutaneously in the dorsal hip with MZ2-MEL (5 × 10^6^ cells/mouse) or SH-SY5Y (20 × 10^6^ cells/mouse) resuspended in 100 μL 10 mg/mL liquid Matrigel (BD Franklin Lakes, NJ, USA, catalog number 356234). Mice were randomly assigned to receive 3 mg/kg Analog-1 or vehicle only (8% DMSO) (control mice). When tumors were palpable, Analog-1 or vehicle were injected into the tail vein every other day for fourteen days. Tumor volume was calculated using the formula π/6 [w_1_ × (w_2_)^2^], where w_1_ represents the largest tumor diameter and w_2_ represents the smallest tumor diameter. Body weight and general physical status of the animals were recorded daily, and mice were killed by cervical dislocation after being anesthetized with xylazine (Xilor 2%, Bio98 Srl, Milan, Italy), when they showed signs of poor health. The dose of 3 mg/kg was established by a preliminary evaluation of in vivo toxicity of Analog-1 using the procedure reported by the chemical compounds test guidelines number 420 issued by the International Organization for Economic Cooperation and development. The protocol of the latter experiments was approved by the Policlinico San Martino Animal Welfare Body and by the Italian Ministry of Health (Authorization n° 358-2019-PR in data 14 May 2019).

### 2.16. Histology and Immunohistochemistry

Five μm thick sections from formalin-fixed, paraffin-embedded blocks were stained with Ki-67 (Dako, Milan, Italy, catalog number M7240) and Dako Envision System horse radish peroxidase (HRP). Peroxidase activity was detected by a 6–10 min incubation at room temperature with Liquid diaminobenzidine (DAB) Substrate Chromogen System (Dako, catalog number. K3468). Counterstaining was performed with Mayer’s hematoxylin (Sigma, catalog number MHS32). Data were expressed as the mean value of Ki67 positive cells counted in 10 fields ± SEM. TUNEL assay on tumor sections was performed by TUNEL staining with the ApopTag plus Peroxidase in Situ Apoptosis kit (Millipore, Billerica, MA, USA, catalog number S7101) according to the manufacturer’s protocol. Briefly, tissue sections were deparaffinised, permeabilized and stained with a solution containing terminal deoxynucleotidyl transferase and nucleotide mixture for 1 h at 37 °C. After washing, the slides were dried at room temperature and counter-stained with DAPI (Vectashield mounting, Vector Laboratories, CA, USA, catalog number H-1200-10). Data were expressed as the mean fluorescence intensity ± SEM of 80 different fields acquired by Opera Phenix high content system at 20× magnification.

### 2.17. Statistical Analysis

Statistical parameters including the exact value of *n* and statistical significance are reported in the figure legends. Results were analyzed using an unpaired *t* Test and ANOVA test where indicated using GraphPad Prism 3.0 software (GraphPad Software, El Camino Real, San Diego, CA, USA). *p* value reported in survival curves was calculated with log-rank test corrected with Bonferroni method. Asterisks indicate statistical significance (*, *p* < 0.05; **, *p* < 0.01; ***, *p* < 0.001, ****, *p* < 0.0001).

## 3. Results

### 3.1. Low RNF5 Expression Determines a Poor Prognostic Outcome in Neuroblastoma and Melanoma Patients 

We analyzed *RNF5* gene expression in the publicly available dataset GSE49710 on R2 Platform, which contains the expression profiles of 498 NB patients. The relationship between *RNF5* expression and disease course in terms of both overall survival (OS) and event free survival (EFS) was analyzed by Kaplan-Meier analysis, considering a follow up of 216 months and selecting the median value as cut-off for defining high or low *RNF5* gene expression. According to such parameters, high *RNF5* expression was associated with a better clinical outcome (*n* = 249) compared to its lower expression observed in 249 cases, who were characterized by a poor prognosis (*p* < 0.001) (Figure 1a). Similarly, high *RNF5* expression was related to an increased event-free survival of NB patients (*n* = 249), whereas low *RNF5* levels were observed for subjects (*n* = 249) with a reduced event-free survival (*p* < 0.01) (Figure 1b). Hence, *RNF5* gene expression is inversely related to outcome in NB. This was also evident when comparing its expression levels in high-risk (HR) and low-risk (LR) patients. HR-NB cases showed a significant downregulation (Log_2_FC = −1.02, ANOVA *p* < 0.0001) of *RNF5* compared to LR-NB ones (Figure 1c). We also assessed the association between RNF5 expression levels and overall survival of melanoma patients included in the publicly available dataset GSE65904. As for the previous analysis, the median cut-off method was used to determine high or low gene expression. Kaplan-Meier curves showed a trend toward lower *RNF5* expression in patients with poor prognosis, which, however, was not significant (*p* > 0.05) (Figure 1d). Therefore, despite the low *RNF5* expression observed also in melanoma patients with poor clinical outcome, the lack of significance did not allow to confirm its prognostic impact in melanoma tumors.

### 3.2. RNF5 Is Expressed by Neuroblastoma and Melanoma Cell Lines and Its Activation by Analog-1 Reduces Tumor Cell Viability

To confirm that RNF5 is a candidate target for the treatment of NB and melanoma, we evaluated its expression in a panel of human NB (SH-SY5Y, SK-N-SH, IMR-32, HTLA-230) and human melanoma (MZ2-MEL, A2058, G-361, SK-MEL-28) cell lines by western blot.

In Western Blot, RNF5 can be detected as either one or two bands at approximately 20 KDa [30]. As shown in Figure 2a, RNF5 was expressed with variable intensity by all the tumor cell lines tested. As positive control, being RNF5 a known CFTR channel regulator, we verified also RNF5 expression in primary human bronchial epithelial (HBE) cells from a non-CF patient (donor ID: HBE121), primary human bronchial epithelial cells from a CF subject (F508del/F508del) (donor ID: HBE55), in immortalized human bronchial epithelial cells overexpressing the F508del-CFTR mutant (CFBE41o-) and for comparison human fibroblasts (Figure 2a, Appendix A). Interestingly, HBE cells derived from CF patients displayed increased RNF5 expression as compared to HBE cells derived from non-CF subjects. This finding is in agreement with the upregulation of RNF5 observed in CF bronchial tissue [31]. The identity of the putative RNF5 band was confirmed by performing a western blot analysis on cells in which RNF5 expression was knocked down by using an RNF5-targeting siRNA (Appendix A).

We next evaluated the effect of small drug-like compounds, recently defined as inhibitors and activators of RNF5, in two representative NB and melanoma cell lines, SH-SY5Y and MZ2-MEL, respectively [8]. Both the cell lines were treated with 10 μM RNF5 activator Analog-1 or vehicle (control) for 24 and 48 h. We choose the dose of 10 μM Analog-1 since this concentration yielded the biological effects resembling activation of RNF5 in HBE cells [8]. Figure 2b shows that Analog-1 significantly reduced tumor cell viability in a time-dependent manner. Specifically, Analog-1 decreased the cell viability of SH-SY5Y by 26% and 42% after 24 and 48 h of treatment, respectively (*p* < 0.01 at 24 and 48 h) and that of MZ2-MEL by 21% and 42% (*p* < 0.05 at 24 h; *p* < 0.0001 at 48 h). In contrast, the same dose of Analog-1 was totally ineffective in normal human fibroblasts (Appendix A). Next, we evaluated the effect of RNF5 inhibitor Inh-2 (10 μM) in the same tumor cell lines. Again, the dose was chosen since this concentration inhibited RNF5 in HBE cells [8]. Inh-2 treatment did not affect NB and melanoma cell viability at 24 and 48 h (Figure 2c). In order to evaluate whether Analog-1 increased protein expression, we performed a western blot on SH-SY5Y and MZ2-MEL cells treated with the drug or vehicle alone. As shown in Figure 2d and Appendix A, RNF5 protein expression was superimposable regardless of treatment with Analog-1, suggesting that the drug modulated RNF5 activity but not its level of protein expression.

### 3.3. Analog-1 Modulates the Metabolism of Neuroblastoma and Melanoma Cell Lines

Given the previous indication that RNF5 promoted ubiquitination and degradation of glutamine transporters, which are aberrantly folded following chemotherapy-induced ER stress, we investigated the effect of Analog-1 on glutamine metabolism of NB and melanoma cell lines [13]. SH-SY5Y and MZ2-MEL were treated with 10 μM Analog-1 or vehicle (control) for 24 and 48 h and evaluated for their glutamine and glutamate intracellular content by spectrophotometrically commercial kit.

Figure 3a shows that Analog-1 decreased the intracellular glutamine content of SH-SY5Y by 26% (*p* < 0.0001) after 24 h and by 50% after 48 h (*p* < 0.0001) in comparison to control cells. Accordingly, Analog-1 also reduced the intracellular glutamate content of SH-SY5Y by 55% at 24 h (*p* < 0.0001) and by 81% at 48 h (*p* < 0.0001) in comparison to control cells (Figure 3b). Similar results were obtained in MZ2-MEL where Analog-1 decreased the intracellular glutamine level by 27% at 24 h (*p* < 0.0001) and by 43% at 48 h (*p* < 0.0001) and the intracellular glutamate level by 71% at 24 h (*p* < 0.0001) and by 83% at 48 h (*p* < 0.0001) in comparison to control cells (Figure 3a,b).

Next, we investigated whether Analog-1 modulated the energetic metabolism by evaluating its effect on the activity of ATP-synthase, which is required for malignant tumor growth and was recently proposed as an emerging target for cancer therapy [32,33]. For this purpose, we evaluated the ATP-synthase activity in SH-SY5Y and MZ2-MEL cell lines treated with 10 μM Analog-1 or vehicle alone (control) for 48 h by luminometric measurements. As shown in Figure 4a, Analog-1 led to a decrease in ATP synthesis of 43% in SH-SY5Y (*p* < 0.001) and of 27% in MZ2-MEL (*p* < 0.01) cell lines in comparison to control cells.

The reduction in mitochondrial ATP-synthesis can be compensated by enhanced glycolytic activity that, though less efficient, also produces ATP [34]. Therefore, we analyzed the activity of the key glycolytic enzymes including hexokinase (HK), phosphofructokinase (PFK), pyruvate kinase (PK), and lactate dehydrogenase (LDH) in SH-SY-5Y and MZ2-MEL treated with 10 μM Analog-1 or vehicle (control) for 48 h, by spectrophotometric assays. Figure 4b–e shows that Analog-1 significantly increased the activity of all the glycolytic enzymes in both NB and melanoma cell lines in comparison to control cells. Specifically, HK was increased by 323% in SH-SY-5Y (*p* < 0.0001) and by 200% in MZ2-MEL (*p* < 0.0001) (Figure 4b), PFK was increased by 181% in SH-SY5Y (*p* < 0.0001) and by 237% in MZ2-MEL (*p* < 0.0001) (Figure 4c), PK was increased by 160% in SH-SY5Y (*p* < 0.0001) and by 176% in MZ2-MEL (*p* < 0.0001) (Figure 4d), and LDH was increased by 134% in SH-SY5Y (*p* < 0.0001) and by 137% in MZ2-MEL (*p* < 0.0001) (Figure 4e). However, despite the enhancement of lactate fermentation, the ATP/AMP ratio, a good indicator of cellular energy status, was dramatically reduced by 70% in Analog-1-treated- SH-SY5Y (*p* < 0.0001) and by 60% in Analog-1-treated-MZ2-MEL cells (*p* < 0.0001) in comparison to the control counterpart (Figure 4f), suggesting that the inhibition of aerobic metabolism is not counteracted by the anaerobic glycolysis increment.

Taken together, these data indicated that the RNF5 activator Analog-1 affected tumor cells through down modulation of glutamine availability and energy metabolism. This latter event led the cancer cells to try to compensate the low content ATP by upregulating glycolytic enzymes, without reaching a complete recovery.

### 3.4. Analog-1 Induces Apoptosis by Promoting Oxidative Stress and Decreases the Proliferation of Neuroblastoma and Melanoma Cells

Given that inhibition of the ATP-synthase promotes an increase in the mitochondrial membrane potential (Δψm) and the subsequent increase in superoxide radical production, we investigated the effect of Analog-1 on oxidative stress of NB and melanoma cell lines [35]. SH-SY5Y and MZ2-MEL were treated with 10 μM Analog-1 or vehicle (control) and evaluated for ROS content for up to 180 min by time-lapse immunofluorescence.

As shown in Figure 5a, in SH-SY5Y Analog-1 increased the oxidative stress by 132% after 60 min (*p* < 0.05), by 142% after 120 min (*p* < 0.05), and by 166% after 180 min (*p* < 0.01) in comparison to control cells. The Analog-1-induced oxidative stress was superimposable to that promoted by phorbol-miristate-acetate (PMA) vs. control (after 60 min *p* < 0.01; after 120 min *p* < 0.01 and after 180 min *p* < 0.001). In MZ2-MEL Analog-1 increased the oxidative stress by 211% after 120 min (*p* < 0.05) and by 340% at 180 min (*p* < 0.0001) in comparison to control cells. Even in MZ2-MEL the PMA-induced oxidative stress was similar to that promoted by Analog-1 at 120 min (*p* < 0.001) and less pronounced at 180 min (*p* < 0.0001).

Since oxidative stress is often associated to cell death induction, we evaluated the effect of 10 μM Analog-1 on apoptosis of NB and melanoma cell lines by immunofluorescence [36]. Figure 5b shows that both SH-SY5Y and MZ2-MEL cells presented a significant increase in the percentage of apoptotic cells in a time- and dose-dependent manner. As a positive control, we treated NB and melanoma cells lined with MG-132, a known proteasome inhibitor, which has been shown to induce apoptosis in tumor cells [37].

Next, we evaluated the effect of 10 μM Analog-1 on the proliferation of the same cell lines at different time points. Figure 4c shows that Analog-1 significantly reduced the proliferation of SH-SY5Y starting from 72 h (*p* < 0.01) to 144 h (*p* < 0.001). An earlier anti-proliferative effect, starting from 24 h (*p* < 0.05) up to 144 h (*p* < 0.0001), was observed in MZ2-MEL cells (Figure 5c). In a set of experiments, we also evaluated the morphology of NB and melanoma cells treated with Analog-1 or vehicle (control) for 24 and 48 h by analyzing cell roundness. As shown in Figure 5d treatment of SH-SY5Y and MZ2-MEL with Analog-1 for 48 h caused slight albeit significant morphological changes (*p* < 0.0001 for SH-SY5Y and *p* < 0.0001 for MZ2-MEL).

Finally, Analog-1 did not exert any effect on tumor cell autophagy (Appendix A) neither in presence of the autophagy inhibitor SAR-405 nor in presence of the autophagy activator Torin-1.

### 3.5. Analog-1 Delays Tumor Growth in Human Neuroblastoma and Melanoma Animal Models

To evaluate the ability of Analog-1 to inhibit in vivo tumor growth, female nude mice were subcutaneously inoculated with SH-SY5Y (20 × 10^6^/mouse) and with MZ2-MEL (5 × 10^6^/mouse) cells. Tumors were allowed to grow to a palpable volume before the mice were randomized into two groups (6 mice per group) receiving intravenous administration of Analog-1 (3 mg/kg body weight) or vehicle (DMSO) (control group) every other day for 14 days. We choose the dose of 3 mg/kg since this concentration was shown to be the maximal tolerated dose in preliminary experiments (data not shown). Tumor growth was monitored by a caliper until the animals were sacrificed due to signs of poor health.

Figure 6 shows that Analog-1 significantly delayed tumor growth in both NB- (Figure 6a) and melanoma-bearing mice (Figure 6b). No differences in body weight or side effects including ruffled fur, vomiting, hyperactivity or loss of ambulation and breathing depression, were observed in mice during treatment with Analog-1 (data not shown). Immunohistochemical analysis of paraffin-embedded tumor sections from NB and melanoma mice reveals that Analog-1 significantly decreased the level of Ki-67, which is a marker of cell proliferation, in comparison to control mice (NB-bearing mice Analog-1 vs. Control *p* < 0.001; melanoma-bearing mice Analog-1 vs. Control *p* < 0.05) (Figure 6c). Furthermore, Analog-1 treatment significantly increased the rate of tumor apoptotic cells in both NB- and melanoma-bearing mice evaluated by TUNEL assay (NB-bearing mice Analog-1 vs. Control *p* < 0.0001; melanoma-bearing mice Analog-1 vs. Control *p* < 0.001) (Figure 6d).

## 4. Discussion

Neuroblastoma (NB) and melanoma are neural crest-derived tumors of childhood and adulthood, respectively, characterized by a high aggressiveness and propensity to metastasize [38]. NB arises from the sympathetic ganglia and adrenal medulla and represents the most common pediatric extracranial malignant tumor. The main first-line therapy for high-risk NB patients includes chemo- and radiotherapies, which nevertheless cause remarkable toxicity, development of drug resistance, and are often associated to relapse and disease progression [39]. Melanoma originates from neural-crest derived melanocytes and represents the most common skin tumor of the adult [40]. Despite the considerable advances in melanoma treatment due to the introduction of targeted therapy and immune check-point inhibitors, many tumors are intrinsically resistant or acquire drug resistance, resulting in disease recurrence or progression [41,42].

In this study, we demonstrated that the ubiquitin E3 ligase RNF5, an endoplasmic reticulum (ER) ubiquitin ligase implicated in ER-associated degradation of misfolded proteins, was expressed in NB and melanoma patients and that its expression positively correlated with better clinical outcome in terms of overall and event-free survival. In parallel, we observed that low-risk NB stages showed higher RNF5 expression in comparison to high-risk disease stages. This evidence points out the potential clinical relevance of *RNF5* gene, which may represent a novel biomarker predicting the outcome of NB. Furthermore, *RNF5* overexpression in high-risk neuroblastoma patients at diagnosis suggest its role as potential druggable target. Albeit the survival analysis did not reach the statistical significance, in low RNF5 levels were associated with a poor clinical outcome also in melanoma patients. These results highlight the potential prognostic value of *RNF5* expression in two different neural crest-derived tumors.

In agreement with our results, breast cancer patients with high expression of RNF5 and low expression of the glutamine transporter SLC1A5 were associated with better prognosis than tumors expressing high SLC1A5 and low RNF5 [13]. In contrast, in acute myeloid leukemia and hepatocellular carcinoma, high expression of RNF5 correlated with poor prognosis [11,43]. From these observations, it is evident that the role of RNF5 in cancer is still controversial and its pro- or anti-tumor activity likely depend on the tested tumor type. In particular, we hypothesize that the activity of RNF5 is related to different factors including oncogenic background, metabolic phenotype, epigenetic features and tumor microenvironment. For instance, neuroblastoma is characterized by the amplification of the oncogene *N-Myc*, which in turn alters mitochondrial metabolism, making tumor cells dependent on exogenous glutamine [44]. In light of this evidence, it is conceivable that RNF5 exerts a significative anti-neuroblastoma effect by reducing the intracellular content of glutamine likely through degradation of its transporters. Analogously, RNF5 activation slows the proliferation of melanoma cells, which, irrespective of their oncogenic background, depend on glutamine to grow [45]. Another important aspect to be considered is that E3 ubiquitin ligases, rather than a direct effect on tumor cells, play a crucial role in regulating anti-tumor immune responses by the degradation of immune checkpoints and the activation of immune-related pathways [46] However, the specific role of RNF5 in the latter phenomena has not yet been defined. Notably, rnf5^−/−^ mice present an enhanced antitumor response, which is coupled with greater recovery and function of CD8^+^ lymphocytes in melanoma tumors [12]. A further assessment of RNF5-dependent regulation of T cells indicated that RNF5 deficiency inhibits functional T cell exhaustion upon chronic antigen exposure, which often occurs in cancer [47]. Considering the complex and extensive activities regulated by ubiquitination, blocking or activating E3 ligases to treat tumors may exert adverse effects on other normal metabolic activities and on the immune system. Thus, exploring and solving this problem remains a considerable challenge.

RNF5 is also involved in the premature degradation of cystic fibrosis transmembrane conductance regulator (CFTR) chloride channel [6], and its genetic or pharmacological inhibition in vivo attenuated the defects associated to deletion of phenylalanine 508 (F508del), which is the most frequent CF mutation impairing CFTR trafficking to the cell surface [8,31] Two novel drugs targeting RNF5 have been discovered by a computational approach, based on ligand docking and virtual screening. The first one, Inh-2, is a drug-like small molecule that inhibits RNF5 while the other one, Analog-1, is a close analog of Inh-2 and behaves as an RNF5 activator [8].

In this study, we evaluate the biological effects of Inh-2 and Analog-1 in NB and melanoma cell lines as well as normal cells expressing RNF5. The specific effect of these drugs were evaluated in previous studies showing that Inh-2 and Analog-1, respectively, reduce and increase the ubiquitination of Autophagy-related (ATG) 4b, a target of RNF5 [8]. Our results demonstrate that Analog-1 specifically inhibited the viability of NB and melanoma cell lines without any cytotoxic effect on fibroblasts, suggesting that activation of RNF5 represents a potential anti-tumor treatment strategy. Interestingly, similar results were obtained on bronchial epithelial cells, where Analog-1 slowed down cell proliferation while inh-2 increased its rate [8]. In contrast, Inh-2 exerted effects neither in NB and melanoma cell lines nor in normal cells. We can speculate that the lack of effect of Inh-2 on NB and melanoma cell lines could be due to their high-proliferation rate, which cannot be further increased by RNF5 inhibition. Of note, the anti-tumor effect of Analog-1 was also demonstrated in in vivo models of human NB and melanoma where the drug delayed tumor growth by reducing proliferation and increasing apoptosis. The potential effect of Analog-1 on the tumor microenvironment including immune cells remains to be investigated in syngeneic models. Importantly, Analog-1, which has never tested before in vivo, was safe and did not cause any sign of toxicity.

In breast cancer, RNF5 is involved in the degradation of glutamine transporters SLC1A5 and SLC38A2, which are aberrantly folded following chemotherapy-induced ER stress [13]. In line with these results, we observed that Analog-1 reduced the intracellular glutamine and glutamate concentrations in both NB and melanoma cells likely through the induction of SLC1A5 and SLC38A2 ubiquitination and subsequent degradation. This is relevant evidence since NB cells rely on SLC1A5 to maintain sufficient levels of glutamine that are essential for proliferation. Moreover, SLC1A5 expression was markedly elevated in high-stage NB tumor samples compared with low-stage ones and it was significantly associated with poor prognosis and low survival of NB patients [48]. SLC1A5 was found highly expressed also in melanoma cells where it promoted cell proliferation and resistance to conventional therapies [49,50]. Accordingly, pharmacologically and genetic inhibitions of SLC1A5 led to regression of cell proliferation and were proposed as a potential therapeutic target for melanoma [50,51].

In this study, we demonstrated that the energetic metabolism of tumor cells is also dramatically downregulated by Analog-1 through the inhibition of ATP-synthase activity, resulting in ATP reduction and AMP increase. ATP-synthase is a central enzyme of the cellular energy metabolism, which is highly expressed during cancer growth and linked to a poor prognosis in almost all tumors [32,52,53,54,55]. Targeting ATP-synthase has been proposed as a promising approach for cancer therapy [33,36]. However, several challenges in translating preclinical ATP synthase targeting drugs to the clinic mainly remain due to their unacceptable in vivo toxicity [56]. In the light of these considerations, for the first time we propose a new drug that exerts a potent inhibitory activity on ATP-synthase without causing any in vivo toxicity.

ATP synthase provides most of the ATP required to maintain cellular activities, however glycolysis also generates ATP, albeit less efficiently than mitochondrial respiration. Interestingly, NB and melanoma cell lines treated with Analog-1 presented higher activity of glycolytic enzymes. We hypothesized that this latter effect was associated with the attempt to compensate for the reduced aerobic ATP synthesis. However, the low ATP/AMP ratio suggested that ATP production was recovered only in part by the anaerobic metabolism.

Inhibition of ATP synthase was also involved in oxidative stress through the generation of reactive oxygen species (ROS) that, in turn, can have pro- and anti-tumoral effects [57,58,59]. On one hand, ROS may drive cell proliferation and invasion, on the other hand, they may induce programmed cell death [59]. With this in mind, we found that Analog-1 increased ROS and induced apoptosis of NB and melanoma cells. In contrast, autophagy, which has been reported to be induced by ROS in cancer [60,61] was not modulated by Analog-1 in NB and melanoma models.

## 5. Conclusions

Despite the advances in melanoma and NB treatment, patients with advanced stage of disease are often refractory to therapy and relapse. Thus, innovative therapeutic approaches to be combined with the conventional ones should be investigated. Here we provide evidence that a novel RNF5 activator, Analog-1, exerted a potent anti-tumor effect in in vitro and in vivo NB and melanoma models through down-regulation of cell metabolism, inhibition of proliferation and induction of apoptosis. Overall these findings validate RNF5 as a drug target for neuroectodermal tumors, and Analog-1 as a promising small drug to progress towards full preclinical development, paving the way to future clinical trials for NB and melanoma patients. As a note of caution, current functional studies will help to define the mechanisms underlying the biological activities exerted by Analog-1 in order to exclude any off-target effect. In addition, the effects on the immune system remain to be determined.

## Figures and Tables

**Figure 1 cancers-14-01802-f001:**
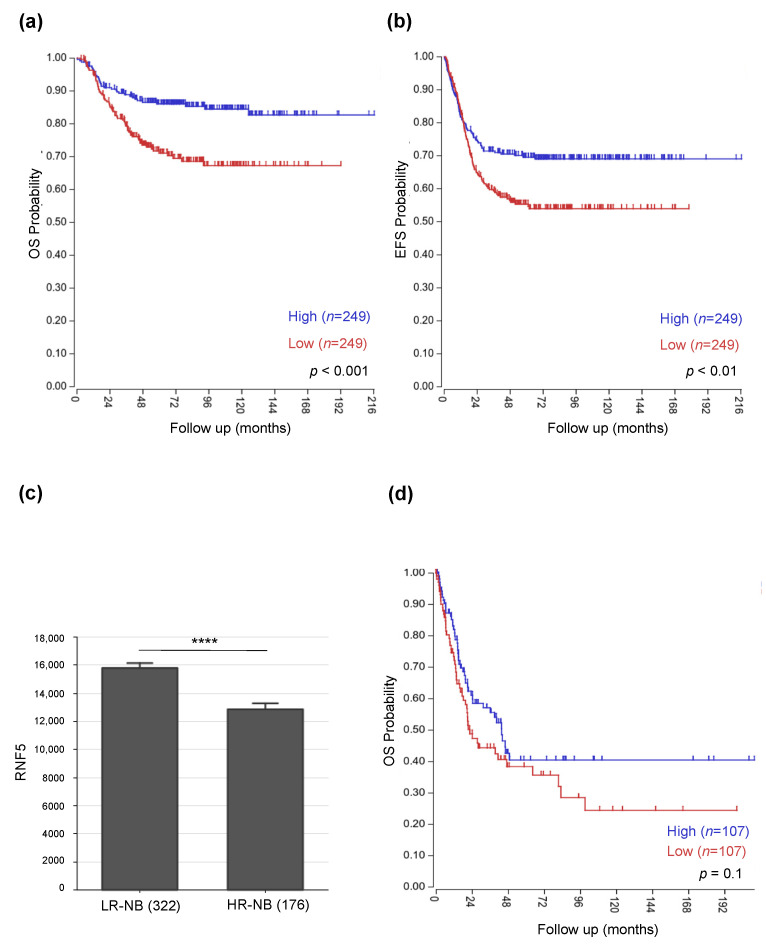
RNF5 expression profile is associated with survival of neuroblastoma and melanoma patients. (**a**,**b**): Kaplan-Meier analysis shows that neuroblastoma patients with lower RNF5 expression levels have a poorer prognostic outcome both in terms of overall survival (OS) (**a**) and event free survival (EFS) (**b**). Red and blue curves refer to RNF5 low and high expression, respectively. Y axis reports OS and EFS probabilities, while the time of follow up, expressed in months, is reported on the X axis. Statistical analysis was performed with log-rank test corrected with Bonferroni method (*p* < 0.01; *p* < 0.001). (**c**): bar graph represents the differential expression of RNF5 in High Risk (HR) and Low Risk (LR) neuroblastoma (NB) patients. Statistical analysis was performed by one way ANOVA test. Asterisks indicate statistical significance (Log2FC= −1.02, **** *p* value <0.0001). (**d**): survival analysis of patients with melanoma shows that a low expression of RNF5 is observed in patients with poor prognosis (*n* = 107 red curve) compared to subjects with favorable clinical outcome (*n* = 107 blue curve), but this association is not significant (*p* value > 0.05). OS probability is reported on Y axis, while the months of follow-up are shown on the X axis. Statistical analysis was performed with log-rank test corrected with Bonferroni method.

**Figure 2 cancers-14-01802-f002:**
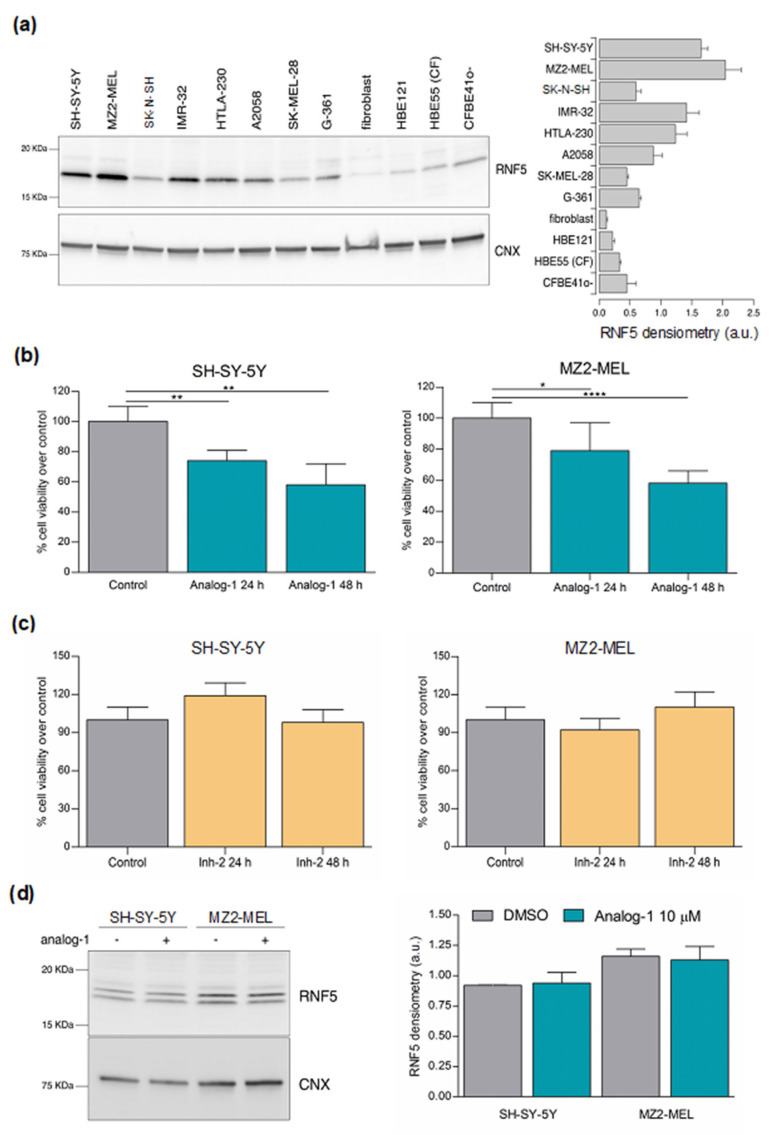
RNF5 activation by Analog−1 decreases tumor cell viability. (**a**): a representative western blot and densitometric analysis of RNF5 (lower band) in human neuroblastoma (SH-SY5Y, SK−N−SH, IMR3−2, HTLA−230), melanoma (MZ2−MEL, A2058, G−361, SK−MEL−28), non small lung cancer (Calu−3) cell lines, human fibroblasts, primary human BE from a non−CF patient (HBE121), primary human BE from a CF subject (HBE55) cells and human immortalized bronchial epithelial (BE) from a cystic fibrosis (CF) patient (CFBE41o-). Calnexin (CNX) was used as housekeeping protein. (**b**): cell viability of SH−SY5Y neuroblastoma and MZ2−MEL melanoma cell lines treated with 10 μM Analog−1 for 24 and 48 h as determined by Trypan Blue Assay. Results are expressed as mean of the percentage of viable cells over control ± SD from three different experiments. Statistical analysis was performed using Unpaired *t* Test. Asterisks indicate statistical significance (*: *p* < 0.05; **: *p* < 0.01; ****: *p* < 0.0001). (**c**): cell viability of SH−SY5Y neuroblastoma and MZ2−MEL melanoma cell lines treated with 10 μM In−h2 as determined by Trypan Blue Assay. Results are expressed as the mean of the percentage of viable cells over control ± SD from three different experiments. (**d**): a representative western blot and densitometric analysis of RNF5 (lower band) in human neuroblastoma SH−SY5Y and human melanoma MZ2−MEL cell lines treated with vehicle alone (DMSO) or with Analog−1 (10 µM) for 24 h.

**Figure 3 cancers-14-01802-f003:**
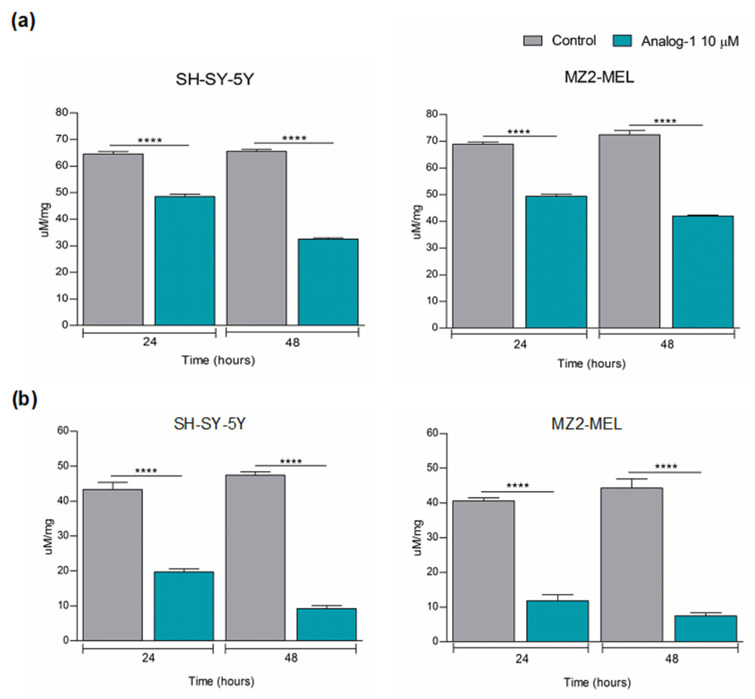
Analog-1 reduces the glutamine and glutamate intracellular content of neuroblastoma and melanoma cells. (**a**,**b**): glutamine (**a**) and glutamate (**b**) intracellular content of neuroblastoma SH-SY5Y and melanoma MZ2-MEL cell lines treated with 10 μM Analog-1 or vehicle for 24 and 48 h and evaluated by spectrophotometric analysis. Data are expressed as the mean ± SD from three different experiments. Statistical analysis was performed using Unpaired *t* Test. Asterisks indicate statistical significance (****: *p* < 0.0001).

**Figure 4 cancers-14-01802-f004:**
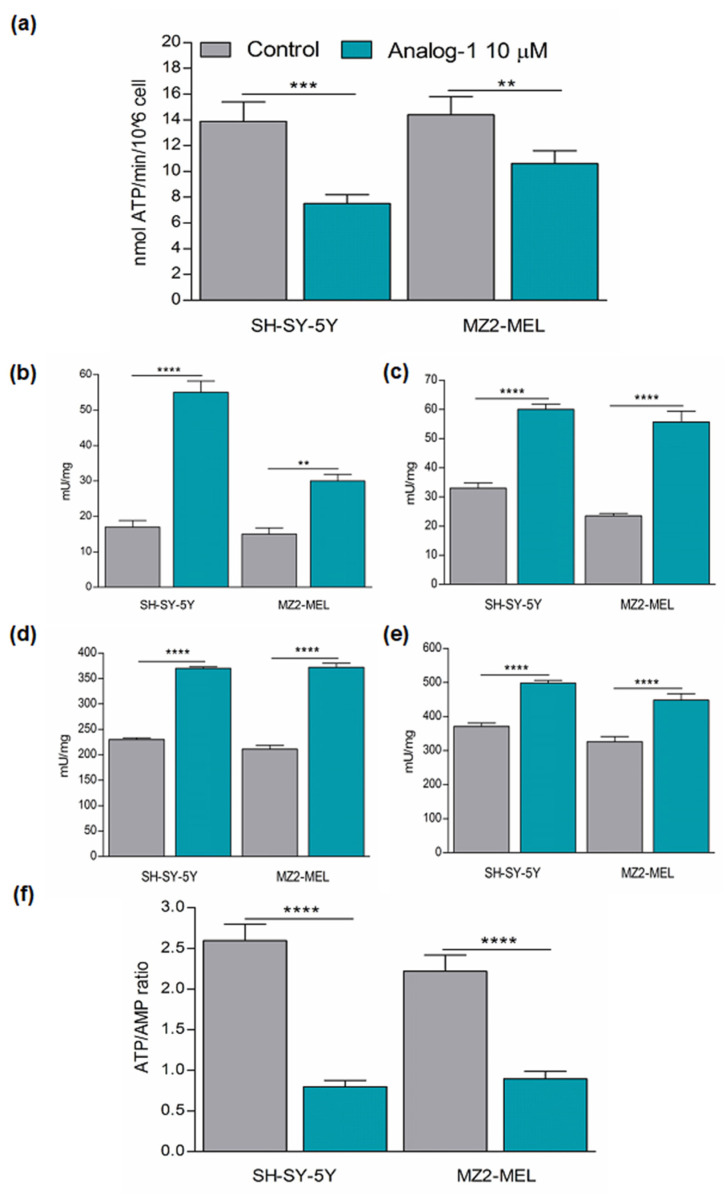
Analog-1 down-modulates the energetic metabolism of neuroblastoma and melanoma cells. (**a**): ATP synthase activity (nmol ATP/min/10^6^ cells) after the addition of 10 mM pyruvate + 5 mM malate and measured by luminometric analysis. (**b**–**e**): activity of Hexokinase (**b**), Phosphofructokinase (**c**), Pyruvate kinase (**d**), Lactate Dehydrogenase (**e**) of neuroblastoma SH-SY-5Y and melanoma MZ2-MEL cell lines untreated or treated with Analog-1 for 24 and 48 h evaluated by spectrophotometric analysis. (**f**): ATP/AMP ratio in neuroblastoma SH-SY5Y and melanoma MZ2-MEL cell lines untreated or treated with Analog-1 for 24 and 48. Each graph is representative of three experiments performed and data are expressed as the mean ± SD. Statistical analysis was performed using Unpaired *t* Test. Asterisks indicate statistical significance (**: *p* < 0.01; ***: *p* < 0.001; ****: *p* < 0.0001).

**Figure 5 cancers-14-01802-f005:**
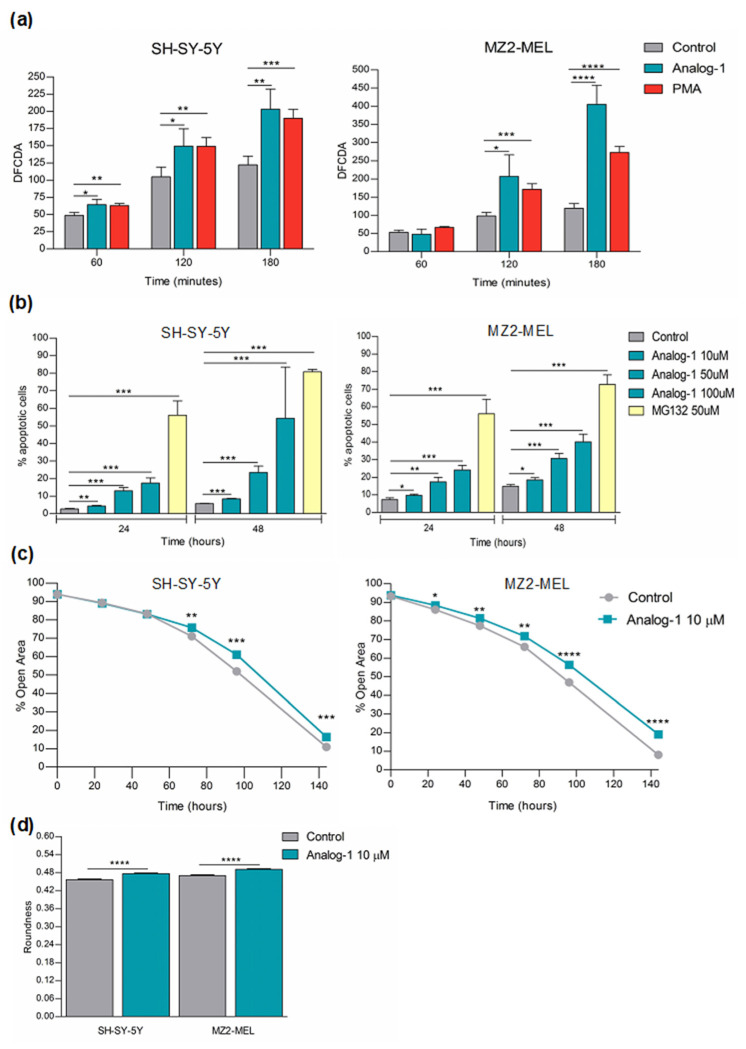
Analog-1 affects reactive oxygen species production, apoptosis, proliferation and morphology of neuroblastoma and melanoma cells. (**a**): oxidation stress of SH-SY5Y and MZ2-MEL cells treated with Analog-1 10 μM or vehicle alone (negative control) or phorbol-miristate-acetate (PMA) (positive control) for different times up to 180 min, evaluated by measuring oxidized dye Dichloro-dihydrofluorescein-diacetate (DCFDA) in living cells by time-lapse imaging of cells using the Opera Phenix high-content screening system. Data are expressed as the mean value of DFCDA intensity ± SEM (*n* = 4). Statistical analysis was performed using Unpaired *t* Test. Asterisks indicate statistical significance (*: *p* < 0.05; **: *p* < 0.01; ***: *p* < 0.001; ****: *p* < 0.0001). (**b**): percentage of apoptotic cells upon treatment of SH-SY5Y and MZ2-MEL cells with Analog-1 10–50–100 μM or vehicle alone (negative control) or the proteasome inhibitor MG-123 (positive control) for 24 and 48 h. The cells were stained with Hoecht 33342 and Propidium Iodide and then imaged by using the Opera Phenix high-content screening system. Data are expressed as the mean of % apoptotic cells ± SEM (*n* = 6). Statistical analysis was performed using Unpaired *t* Test. Asterisks indicate statistical significance (*: *p* < 0.05; **: *p* < 0.01; ***: *p* < 0.001). (**c**): proliferation of SH-SH5Y neuroblastoma and MZ2-MEL melanoma cell lines treated with 10 μM Analog-1 or vehicle alone (negative control) at different time points and evaluated by using the Opera Phenix high-content screening system. Data are expressed as the mean of % open area of 5 different fields ± SEM. Statistical analysis was performed using Unpaired *t* Test. Asterisks indicate statistical significance (*: *p* < 0.05; **: *p* < 0.01; ***: *p* < 0.001; ****: *p* < 0.0001). (**d**): morphological changes of SH-SH5Y neuroblastoma and MZ2-MEL melanoma cell lines treated with 10 μM Analog-1 for 48 h and evaluated by the analysis of cell roundness using Opera Phenix high-content system. Data are expressed as mean of single cell roundness ± SEM (*n* = 3000). Statistical analysis was performed using Unpaired *t* Test. Asterisks indicate statistical significance (****: *p* < 0.0001).

**Figure 6 cancers-14-01802-f006:**
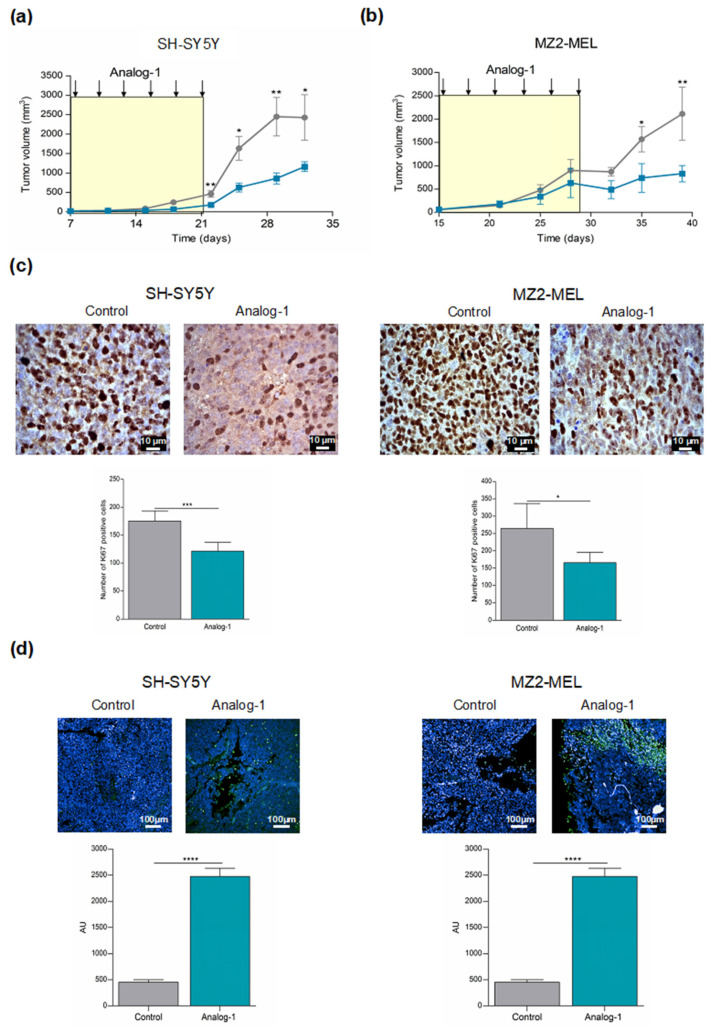
Analog-1 delays in vivo neuroblastoma and melanoma growth. (**a**,**b**): tumor volume of five-week old female nude mice subcutaneously inoculated with 20 × 10^6^ SH-SY5Y (**a**) and 5 × 10^6^ MZ2-MEL (**b**) cell lines. When tumors were palpable, mice (*n* = 8 SH-SY5Y-bearing mice; *n* = 6 MZ2-MEL-bearing mice) were treated every other day by intravenous injection of Analog-1 (3 mg/Kg body weight) (green line) or vehicle alone (control mice: grey line) for 14 days. Tumor volume was calculated using the formula π/6 [w_1_ × (w_2_)^2^], where w1 represents the largest tumor diameter and w2 represents the smallest tumor diameter. Tumor volume is expressed as mean value ± SEM. Statistical analysis was performed Unpaired *t* Test. Asterisks indicate statistical significance (*, *p* < 0.05; **, *p* < 0.01). (**c**): evaluation of proliferation by immunohistochemical staining of paraffin-embedded sections from Analog-1- and vehicle-treated SH-SY5Y and MZ2-MEL bearing mice with anti-Ki-67 cell proliferation marker. Data are expressed as the mean value of Ki67 positive cells counted in 10 fields ± SEM. Original magnification 63×. Statistical analysis was performed Unpaired *t* Test. Asterisks indicate statistical significance (*, *p* < 0.05; ***, *p* < 0.001). (**d**): evaluation of apoptosis by immunofluorescence analysis of paraffin-embedded sections from Analog-1- and vehicle-treated mice using terminal deoxynucleotidyl transferase-mediated dUTP nick end labeling [TUNEL] assay for the detection of apoptosis. Data are expressed as the mean fluorescence intensity ± SEM of 80 different fields acquired by Opera Phenyx high content system at 20× magnification. Statistical analysis was performed Unpaired *t* Test. Asterisks indicate statistical significance (****, *p* < 0.0001).

## Data Availability

Publicly available datasets were analyzed in this study. This data can be found on the R2 Genomics Analysis and Visualization Platform at http://r2.amc.nl (accessed on 1 October 2020).

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
