# Peer review of "Targeting of Ubiquitin E3 Ligase RNF5 as a Novel Therapeutic Strategy in Neuroectodermal Tumors"

_cancers, 2022, doi:10.3390/cancers14071802_

Round 1

Reviewer 1 Report

Principi et al., described their findings about RNF5, a ubiquitin ligase involved in the degradation of misfolded proteins, as an innovative drug target for neuroblastoma and melanoma patients. Authors have examined the therapeutic effect of Analog-1, an RNF5 activator, in vivo neuroblastoma and melanoma models. Further, authors demonstrated that treatment with Analog-1 reduced cell viability and energy metabolism in addition to the marked increase of oxidative stress and significant delay of tumor growth. Overall, the authors observed that low-risk NB stages showed higher RNF5 expression in comparison to high-risk disease stages, suggesting its role as potential druggable target and novel biomarker. While the similar line of studies reported already, the current article has interesting observation which is beneficial to researchers in cancer and drug resistance tumors.

I had no specific suggestions for the authors with exception of the following:

  1. The title of the article does not accurately convey its contents. Title says Targeting of ubiquitin E3 ligase RNF5 as a novel therapeutic 2 strategy in cancer, whereas the study presented only Neuroblastoma and melanoma
  2. Suggested to mention Scale bar (missing or not visible) in all microscopy/staining images.
  3. Suggested to add catalog nos. to all the chemicals/media/antibodies used in methods.
  4. For glutamine and glutamate determination which kit was used is not mentioned.
  5. In siRNA transfection and Cell viability assay how many cells were used is missing.
  6. For histology which sections were fixed?
  7. For WB with RNF5 which band was used for quantification (upper or lower)?
  8. In 498 NB specimens what was the age group/sex of the subjects?

Author Response

We thank the reviewer for his/her relevant comments. The reply to these comments is enlisted below together with the revised manuscript.

  • The title of the article does not accurately convey its contents. Title says Targeting of ubiquitin E3 ligase RNF5 as a novel therapeutic strategy in cancer, whereas the study presented only Neuroblastoma and melanoma.

We agree with the reviewer and we changed the title in “Targeting of ubiquitin E3 ligase RNF5 as a novel therapeutic strategy for neuroectodermal tumors”

  • Suggested to mention Scale bar (missing or not visible) in all microscopy/staining images.

Visible Scale bar have been included  in all microscopic images (Figure 6, panels c and d).

  • Suggested to add catalog nos. to all the chemicals/media/antibodies used in methods.

Catalog numbers of all chemicals/media/antibodies used have been included in the material and method section and are in red font.

  • For glutamine and glutamate determination which kit was used is not mentioned.

We have added the name and catalog number of the glutamine and glutamate kit, as requested (page 5, Material and Methods, “Glutamine and glutamate determination” section).

  • In siRNA transfection and Cell viability assay how many cells were used is missing.

We have added the number of the cells used in the siRNA transfection (2 x106 cells) and in the  cell viability (1,5 x106 cells)  assays, as requested.  Materials and Methods, page 4, “SiRNA transfection” section and “Cell viability” section, respectively.

  • For histology which sections were fixed?

We have now indicated that for histology all tumor sections were fixed with formalin Materials and Methods, page 7.

  • For WB with RNF5 which band was used for quantification (upper or lower)?

For WB with RNF5 the lower band shown in Figure 2, panels a and d, was used for quantification. We have  modified the legend of figure 2, panels a and d, accordingly.

  • In 498 NB specimens what was the age group/sex of the subjects?

The study cohort included 211 female and 287 male patients, with a diagnosis age ranging from 0 to 25 years. Specifically, the 75% of patients received the diagnosis before 3 years of age. This sentence is now included in the text, at page 3, in Materials and Methods “Gene expression analysis” section.

Reviewer 2 Report

The manuscript titled “Targeting of ubiquitin E3 ligase RNF5 as a novel therapeutic 2 strategy in cancer” submitted by Principi and colleagues highlighted an interesting study on ubiquitin ligase RNF5 and its role as a therapeutic target using preclinical models. The authors evaluated the biological effects of Analog-1, a pharmacological activator of RNF5, on cell lines and ultimately its efficacy on suppressing tumor progression in in vivo models. Overall, it is a well-written manuscript which provided some novel information regarding the roles of RNF in tumorigenesis, however, there is a need to further strengthening the mechanistic perspective and the interpretation of the experimental observations prior to publication. The manuscript would be much improved if the author can address these concerns.

Specific comments:

  • In the abstracts, the authors stated that “gene expression profile was evaluated in primary tumors collected at diagnosis using machine learning methods”. However, it was not further discussed, nor elaborated in the manuscript. It is unclear how does these “machine learning methods” play a role in the analysis since the gene expression and survival analysis in figure 1 appears to be derived from conventional approaches of public dataset mining.
  • Figure 2d: RNF5 442 protein expression was not significantly affected by the treatment of Analog-1 and the author concluded that the drug modulated RNF5 activity but not its level of protein expression. While this is in part a valid statement based on the observation, it needs to be further supported by evaluating the downstream pathway of RNF5 including the ubiquitination of target gene. Similarly, for the subsequent evaluation of RNF5 modulated phenotypes by treatment of Analog-1 and Inh-2, it is essential to demonstrated target engagement and an on-target modulation of phenotypes.
  • Figure 3 and 4: Rescue experiments of Analog-1 treatment on RNF5-silenced cells would strengthen the metabolic modulating roles of RNF5 since it is currently unclear if these phenotypic changes are due to the on-target effect of the treatment. In the discussion the authors hypothesize that Analog-1 reduced the intracellular glutamine and glutamate concentrations likely through the induction of SLC1A5 and SLC38A2 ubiquitination and subsequent degradation. To support the oxidation stress hypothesis proposed in the present manuscript, SLC1A5 and SLC38A2 ubiquitination and degradation should be evaluated but not just mentioned.
  • Discussion: I appreciate that the authors provided a comprehensive overview on the current understanding of RNF5. One of the key highlights is that RNF5 seems to play a diverse and context-dependent roles in tumor progression. Because of that, the manuscript would be further improved if the author can put their observation in context. How does the current observations fit in our currently understanding of RNF5, particularly in the area of anti-tumor immunity? Could RNF5-mediated metabolism and stress crosstalk with some of the immune components?

Round 2

Reviewer 2 Report

The author's attempt to address my previous comments is appreciated though no addition experiment has been performed for clarifying their scientific conclusions. With that being said, the authors did try to provide a more objective description of their findings with an editorial approach which is not ideal, but could be considered adequate.